# Split Batch Normalization: Improving Semi-Supervised Learning under Domain Shift

**Michał Zając**
Jagiellonian University
Nomagic
emzajac@gmail.com

**Konrad Żołna**
Jagiellonian University

**Stanisław Jastrzębski**
Jagiellonian University
New York University

## Abstract

Recent work has shown that using unlabeled data in semi-supervised learning is not always beneficial and can even hurt generalization, especially when there is a class mismatch between the unlabeled and labeled examples. We investigate this phenomenon for image classification and many other forms of domain shifts (e.g. salt-and-pepper noise). Our main contribution is showing how to benefit from additional unlabeled data that comes from a shifted distribution in batch-normalized neural networks. We achieve it by simply using separate batch normalization statistics for unlabeled examples. Due to its simplicity, we recommend it as a standard practice.

## 1 Introduction

Deep neural networks (DNNs) rely on large amounts of labeled data to achieve state of the art performance on supervised learning problems such as image classification or speech recognition (Goodfellow et al., 2016). However, labeling data can be prohibitively costly. Consequently, a common research theme is leveraging unlabeled data to improve sample-efficiency of deep networks.

Semi-supervised learning (SSL) is one possible approach (Chapelle et al., 2010). SSL methods can boost performance of DNNs by jointly training on the labeled and unlabeled data (Miyato et al., 2018; Tarvainen & Valpola, 2017).

General assumption in SSL is that the labeled examples comes from the same distribution as the unlabeled examples. It is unclear to what extent we could generally use additional data that is out of distribution. When unsupervised data comes from distribution that is not very distant from supervised data, e.g. from the same dataset but different classes, one could hope to still have SSL performing better than pure supervised solution. However, recently Oliver et al. (2018) have undermined this hope showing that SSL methods are not robust to class mismatch and their performance could be even worse than pure supervised learning.

In this work we investigate and improve SSL techniques' performance in a scenario when unlabeled and labeled examples do not come from the same distribution. We focus on deep neural networks with batch normalization (Ioffe & Szegedy, 2015) for image classification. We verify that indeed in this setting SSL methods hurt from domain shift of unlabeled data.

To alleviate this problem, we propose to compute separately batch-normalization statistics for the unsupervised and supervised data. We experimentally study the effectiveness of this method on a substantially extended version of the setting considered by Oliver et al. (2018) that includes other possible domain shifts such as salt-and-pepper noise or a change in image contrast in the unlabeled data.

## 2 Related Work

Semi-supervised learning is a popular technique to leverage unlabeled data along with (typically a small amount of) labeled data (Chapelle et al., 2010). However, in some cases using unlabeled examples can hurt performance of the model (Gagliardi Cozman et al., 2003; Oliver et al., 2018).

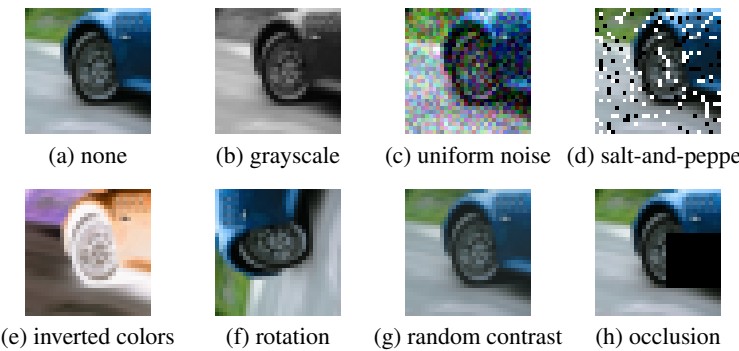

|           |               |                  |                       |
|-----------|---------------|------------------|-----------------------|
| (a) none  | (b) grayscale | (c) uniform noise | (d) salt-and-pepper  |
| (e) inverted colors | (f) rotation | (g) random contrast | (h) occlusion |

Figure 1: Distortions used in the paper to introduce a domain shift between the labeled and unlabeled data.

While there are many approaches to SSL, most of them implicitly or explicitly assume the unlabeled examples follow the distribution of the supervised dataset (Liu et al., 2018). As highlighted by Oliver et al. (2018), a setting in which unlabeled examples follow a different distribution is heavily under-researched. Some works, for example Laine & Aila (2016), consider in their experiments unlabeled data that is out-of-distribution but do not address or analyze the problem.

The most related works are Oliver et al. (2018), which observes that SSL performance can degrade substantially when the unlabeled dataset contains out-of-distribution examples, and Liu et al. (2018), which similarly to us analyzes robustness of SSL techniques. Liu et al. (2018) considers a scenario in which there is a domain shift coming from labels that are missing not completely at random, which can lead to a mismatch in feature distribution. Importantly, the setting we consider is more general – we allow for a systematic domain shift independent of the labeling process. Further, our Split-BN is complementary to the approach of Liu et al. (2018).

Recently Kalayeh & Shah (2019) and Deecke et al. (2019) investigated batch normalization variants that internally model the data using a multimodal distribution. In particular, Deecke et al. (2019) has demonstrated improvements for multi-task supervised learning, a setting related to ours. We believe that it may be interesting direction to apply their method to SSL with domain shift.

## 3 EXPERIMENTS

One of the key claims in Oliver et al. (2018) is that performance of SSL techniques can degrade drastically when the unlabeled data contains a different distribution of classes than the labeled data. We first confirm that a similar phenomenon happens on a wide range of other possible domain shifts in image domain on the CIFAR-10 dataset (see Figure 1 for examples). However, a similar analysis on the more challenging ImageNet dataset reveals that this phenomenon might not be fully general.

Next, we propose and analyse a technique to improve SSL under domain shift. In particular we re-examine our extended setting and observe that in most cases our technique cancels the performance gap between supervised learning and SSL with misaligned data.

### 3.1 EXPERIMENTAL SETTING

Our experiments largely base on Oliver et al. (2018). In particular, we use the same strong Wide ResNet model (Zagoruyko & Komodakis, 2016). We perform experiments with two state-of-the-art SSL methods: Mean Teacher (MT) by Tarvainen & Valpola (2017) and Virtual Adversarial Training (VAT) by Miyato et al. (2018). In all experiments we use 400 labels per class.

In addition to the class mismatch from Oliver et al. (2018), we use popular image distortions (e.g. salt-and-pepper noise) to introduce a systematic difference between the labeled and unlabeled examples.In total, we study 8 different domain shift scenarios: the first based on a class mismatch, and the rest based on applying a fixed type of image distortion to all images in the unlabeled dataset. These distortions are meant to represent challenge present in real world applications. See Figure 1 for visualization. For a detailed description of the transformations, please refer to Appendix A.

Table 1: Test accuracy under the class-mismatch from Oliver et al. (2018), on the CIFAR-10 dataset. Columns correspond to various level of class mismatch between labeled and unlabeled data – from 0% (no mismatch) to 100% (no shared classes). Our Split-BN improves upon the results reported in Oliver et al. (2018), and in particular removes most of the gap between SSL with class mismatch and pure supervised training.

| | 0% | 25% | 50% | 75% | 100% |
|---|---|---|---|---|---|
| **Supervised** | | | 77.0 ± 0.4 | | |
| **Mean Teacher** | **77.7 ± 0.5** | 75.5 ± 0.7 | 74.8 ± 0.3 | 74.5 ± 0.1 | 73.9 ± 0.4 |
| **Mean Teacher + Split-BN** | 77.1 ± 0.2 | **77.6 ± 0.1** | **76.8 ± 0.1** | **77.1 ± 0.4** | **76.2 ± 0.5** |
| **VAT** | **79.3 ± 0.3** | **76.3 ± 0.5** | **75.7 ± 0.5** | 73.8 ± 0.1 | 72.6 ± 0.2 |
| **VAT + Split-BN** | 76.6 ± 0.3 | **76.6 ± 0.4** | 76.3 ± 1.0 | **76.1 ± 0.0** | **76.6 ± 0.5** |

Table 2: Test accuracy on the class-mismatch setting on the ImageNet. Columns correspond to various level of class mismatch between labeled and unlabeled data – from 0% (no mismatch) to 100% (no shared classes).

| | 0% | 25% | 50% | 75% | 100% |
|---|---|---|---|---|---|
| **Supervised** | | | 61.52 ± 1.21 | | |
| **MT** | **65.9 ± 0.8** | **67.2 ± 1.0** | 65.0 ± 0.6 | 63.8 ± 1.6 | 64.0 ± 1.3 |
| **MT + Split-BN** | 65.5 ± 1.2 | 65.8 ± 0.9 | **66.0 ± 0.0** | **65.5 ± 1.1** | **64.1 ± 0.8** |

For simplicity, we fix most of the hyperparameter values to the ones used in Oliver et al. (2018) CIFAR-10 experiments. For the detailed description, please refer to Appendix B. The code to reproduce our experiments will be released upon publication.

## 3.2 SPLIT BATCH NORMALIZATION

Our main contribution is Split Batch Normalization (Split-BN), a technique to improve SSL under domain mismatch, applicable to deep neural networks. Inspired by the simple approach of Li et al. (2016) to domain adaptation, we propose to *compute separately batch-normalization statistics for the unsupervised and supervised dataset*. Note that during test time statistics from supervised data are used.

First, we investigate the class mismatch setting on the CIFAR-10 dataset, as discussed in Oliver et al. (2018). Table 1 reports the results. We can see that in some cases baseline performance is even 3% below accuracy achieved by pure supervised training that does not use additional unsupervised data. Critically, using Split-BN cancels most of the performance degradation in these cases.

To further study class mismatch, we run experiments on the more challenging ImageNet dataset (Deng et al., 2009) rescaled to 32x32 (as in Chrabaszcz et al. (2017)), from which we select 20 random classes as the supervised dataset. Additional unlabeled data comes from a distribution with a varying class mismatch with the labeled data. Results are reported in Table 2. We note that

Table 3: Test accuracy for each SSL technique under various forms of domain shift, on the CIFAR-10 dataset. SSL methods are generally not robust to domain shift. Our Split-BN in most cases leads to an improvement over both SSL and pure supervised training.

| | None | Grayscale | Uniform noise | Salt-and-pepper | Inverted colors | Rotation 90° | Random contrast | Occlusion |
|---|---|---|---|---|---|---|---|---|
| **Supervised** | | | | 79.4 ± 0.1 | | | | |
| **MT** | **83.2 ± 0.1** | 79.3 ± 0.3 | **79.4 ± 0.4** | 78.2 ± 0.4 | 75.2 ± 0.2 | 75.9 ± 0.1 | 80.9 ± 0.0 | 78.6 ± 0.5 |
| **MT + Split-BN** | 83.1 ± 0.6 | **80.4 ± 0.0** | 79.4 ± 0.1 | **78.8 ± 0.5** | **79.8 ± 0.6** | **80.1 ± 0.1** | **82.6 ± 0.6** | **79.9 ± 0.0** |
| **VAT** | 85.1 ± 0.0 | 81.4 ± 0.4 | **79.3 ± 0.4** | 78.7 ± 0.7 | **79.0 ± 0.5** | 77.6 ± 0.4 | 82.0 ± 0.6 | **78.4 ± 0.2** |
| **VAT + Split-BN** | **85.2 ± 0.1** | **83.3 ± 0.1** | 78.2 ± 0.5 | **79.4 ± 0.8** | 78.8 ± 0.2 | **78.6 ± 0.0** | **83.9 ± 0.2** | 80.0 ± 1.4 |

Table 4: Repeated experiments from Table 3 using a model without batch normalization layers. Here also we can see a visible performance drop of SSL under domain shift.

| | None | Grayscale | Uniform noise | Salt-and-pepper | Inverted colors | Rotation 90° | Random contrast | Occlusion |
|---|---|---|---|---|---|---|---|---|
| **Supervised** | | | | $73.1 \pm 0.7$ | | | | |
| **MT** | $77.5 \pm 0.2$ | $73.8 \pm 0.4$ | $73.4 \pm 0.1$ | $73.2 \pm 0.1$ | $73.2 \pm 0.2$ | $72.8 \pm 0.5$ | $76.2 \pm 0.2$ | $73.7 \pm 0.4$ |
| **VAT** | $74.5 \pm 0.4$ | $73.5 \pm 0.2$ | $73.7 \pm 0.1$ | $74.2 \pm 0.4$ | $69.9 \pm 0.2$ | $70.0 \pm 0.5$ | $74.5 \pm 0.5$ | $73.1 \pm 0.5$ |

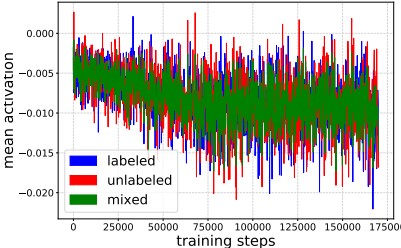 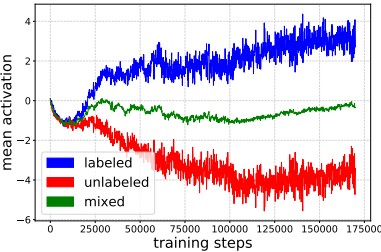

Figure 2: Means of activations before the first batch normalization layer (left), and the last batch normalization layer (right). It can be seen that the means differ in the late layer. Note different ranges on $y$ axes.

in contrast to the previous experiment, on the ImageNet dataset in all cases there is an apparent improvement over the pure supervised case and our Split-BN performs comparably to regular SSL.

Finally, we re-evaluate Oliver et al. (2018) claim using other forms of domain shift applied to the unlabeled data, and test how Split-BN generalizes to this new setting. Table 3 summarizes the results. In all cases with domain shift we observe a large gap in performance, typically between $1\%$ and $4\%$ accuracy drop. Hence, the phenomenon reported by Oliver et al. (2018) is further confirmed and SSL methods indeed are not designed for a domain shift case. Additionally, Split-BN typically cancels most of the negative effect resulting from the domain mismatch, and in many cases leads to a large improvement in performance.

## 3.3 IS BATCH NORMALIZATION THE CULPRIT?

A natural question is whether the lack of robustness of SSL methods simply comes from using batch normalization layers. We repeat the experiments from Section 3.2 using a model from Miyato et al. (2018) and Tarvainen & Valpola (2017), where we removed all batch normalization layers. See Appendix B for experimental details.

Table 4 reports the results. We found that SSL performance may degrade under domain shift also in the case of architectures without batch normalization. This suggests that batch normalization is not the root cause of instability of SSL learning methods.

## 3.4 ACTIVATIONS ANALYSIS

In this section we shed light on how Split-BN improves performance by visualizing the distributions of hidden activations during training. We analyse one of the experiments in Section 3.2 corresponding to training with Mean Teacher and a domain shift introduced by a rotation.

We compute the means of the activations in the layers before the first and the last batch normalization layers, for three sets of examples: (i) labeled examples, (ii) unlabeled examples, and (iii) the whole dataset. Results are shown in Figure 2, and additionally we report evolution of standard deviations in Appendix C. Crucially, the means of the labeled examples and the unlabeled examples become markedly different before the last batch normalization. This suggests that one of the key mechanisms by which Split-BN improves performance is by bringing closer the two distributions.

## 4 CONCLUSIONS

In this work we studied how to use semi-supervised learning if the unlabeled and labeled examples do not come from the same distributions. We provide a simple recommendation: *if you are using a neural network with batch normalization layers, you should compute the normalization statistics separately for the unlabeled and labeled data.* This leads to significant gains in performance; typically canceling out the negative effect of domain shift.

ACKNOWLEDGMENTS

Michał Zając is co-financed by National Centre for Research and Development as a part EU supported Smart Growth Operational Programme 2014-20120, project number POIR.01.01.01-00-0392/17-00. Konrad Żołna is supported by the National Science Center, Poland (2017/27/N/ST6/00828, 2018/28/T/ST6/00211). Stanisław Jastrzębski was supported by grant No. DI 2014/016644 from Ministry of Science and Higher Education, Poland, and No. 2017/24/T/ST6/00487 from National Science Center, Poland.

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

## A    DISTORTION DETAILS

Before applying distortion, we assume pixels are in range $[0, 1]$. As depicted in Figure 1, we used the following distortions:

(a) none;

(b) grayscale: use `tf.image.rgb_to_grayscale` and then stack 3 times to have the same input size;

(c) uniform noise: add uniform noise in range $[-0.2, 0.2]$ and then clip the values back to $[0, 1]$;

(d) salt-and-pepper: every pixel becomes white with probability $10\%$, black with probability $10\%$, and stays the same otherwise;

(e) inverted colors: every color value goes from $x$ to $1 - x$;

(f) rotation: the whole image is rotated by $90°$ counterclockwise;

(g) random contrast: contrast is changed to a random value taken uniformly from $[0.2, 0.8]$;

(h) occlusion: a black 14x14 square is placed on top of the picture at a random location.

## B    DETAILS OF EXPERIMENTAL SETTING

We largely use implementation of the architecture and SSL methods from Oliver et al. (2018).

In all experiments we use the following: batch size equal to 100; Adam optimizer with all hyperparameters but learning rate set to default. In supervised data, there is 400 images per class. Every number we report is accuracy on test set computed using checkpoint with the highest validation accuracy during the training. Every experiment is repeated 2 times.

In experiments from Tables 1, 2, 3, we use Wide ResNet architecture (WRN-28-2 variant with depth 28 and width 2); all Mean Teacher and VAT parameters as in Oliver et al. (2018).

**Experiments from Table 1**    We use the experimental setting from Oliver et al. (2018). The classification is performed on 6 animal classes from CIFAR-10. Additional unsupervised data contains 4 classes and has a varied degree of class mismatch with supervised data – from $0\%$ (no mismatch) to $100\%$ (completely different classes). CIFAR-10 data is preprocessed using global contrast normalization and ZCA normalization. Data is augmented with random horizontal flips, random translation by up to 2 pixels, and Gaussian input noise with standard deviation 0.15. Number of steps is 500000. Learning rate is as in Oliver et al. (2018): for Mean Teacher, initial learning rate is 0.0004, decaying by 0.2 after 400000 steps; for VAT, initial learning rate is 0.003, decaying by 0.2 after 400000 steps. Auxiliary SSL loss is warmed-up for 200000 steps.

**Experiments from Table 2**    The classification is performed on 20 random classes from ImageNet rescaled do 32x32. Additional unsupervised data contains 20 classes and has a varied degree of class mismatch with supervised data – from $0\%$ (no mismatch) to $100\%$ (completely different classes). Data is augmented with random horizontal flips. Learning rate is as in Oliver et al. (2018): initial learning rate is 0.0004, decaying by 0.2 after 400000 steps. Auxiliary SSL loss is warmed-up for 200000 steps.

Table 5: ConvNet architecture, as used in Miyato et al. (2018) and Tarvainen & Valpola (2017), but without batch normalization.

| |
| --- |
| convolutional, 128 filters, $3 \times 3$, same padding, leaky ReLU |
| convolutional, 128 filters, $3 \times 3$, same padding, leaky ReLU |
| convolutional, 128 filters, $3 \times 3$, same padding, leaky ReLU |
| max pooling $2 \times 2$ |
| dropout $p = 0.5$ |
| convolutional, 256 filters, $3 \times 3$, same padding, leaky ReLU |
| convolutional, 256 filters, $3 \times 3$, same padding, leaky ReLU |
| convolutional, 256 filters, $3 \times 3$, same padding, leaky ReLU |
| max pooling $2 \times 2$ |
| dropout $p = 0.5$ |
| convolutional, 512 filters, $3 \times 3$, valid padding, leaky ReLU |
| convolutional, 256 filters, $1 \times 1$, same padding, leaky ReLU |
| convolutional, 128 filters, $1 \times 1$, same padding, leaky ReLU |
| average pooling $(6 \times 6 \to 1 \times 1)$ |
| fully connected, $128 \to 10$ |
| softmax |

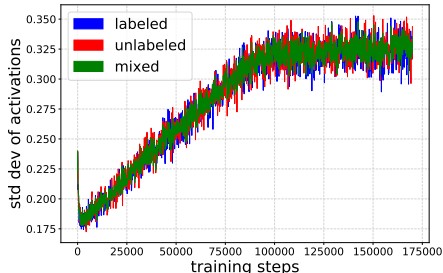 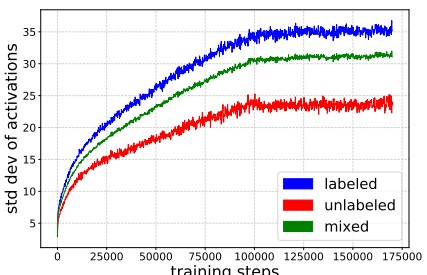

Figure 3: Standard deviations of activations at the early (before first BN) and late (before last BN) layer. It can be seen that the statistics diverge in the late layer. Note different ranges on $y$ axes.

**Experiments from Table 3** Here we use CIFAR-10 without preprocessing. Data is augmented with random horizontal flips and random translation by up to 2 pixels. For each experiment, we search for optimal learning rate in the set $\{0.0003, 0.001, 0.003, 0.01\}$. The training lasts for 170000 steps. Learning rate decays by 0.2 after 100000 steps. Auxiliary SSL loss coefficient is warmed-up for 50000 steps to reach 8.0 for Mean Teacher and 0.3 for VAT.

**Experiments from Table 4** We use ConvNet architecture from Miyato et al. (2018) and Tarvainen & Valpola (2017) with batch normalization eliminated – see Table 5 for details. We use CIFAR-10 without preprocessing. Data is augmented with gaussian noise with standard deviation 0.15, random horizontal flips and random translation by up to 2 pixels. For each experiment, we search for optimal learning rate in the set $\{0.0001, 0.0003, 0.001\}$. The training lasts for 220000 steps. Learning rate decays by 0.25 after every 80000 steps. Auxiliary SSL loss coefficient is warmed-up for 50000 steps to reach 8.0 for Mean Teacher and 0.1 for VAT.

## C PLOTS OF ACTIVATIONS' STANDARD DEVIATIONS

We present here Figure 3 as a supplementary material for the analysis from the subsection 3.4.

