# OpenReview forum: "Split Batch Normalization: Improving Semi-Supervised Learning under Domain Shift"
_ICLR.cc/2019/Workshop/LLD — LLD 2019_

### Official Review · AnonReviewer2 · 2019-04-09
**An worthwhile observation that has its place in the workshop**

**Rating:** 4
**Confidence:** 2

**Review:**

This work proposes an approach to tackle the domain adaptation problem in semi-supervised learning, based on a decoupling of the computation of the batch statistics in the batch normalization layers.

In a setting where the unlabeled data does not follow the supervised data distribution, semi-supervised learning techniques can lead to a degradation of performance with respect to a purely supervised setting. In this work, it is shown that computing the batch normalization statistics separately for the unsupervised and for the supervised data can alleviate the domain shift and lead to improved semi-supervision.

I have a few questions and remarks:

a) The introduction mentions the problem of the domain shift for the unlabeled data. I would add that it is unclear how one could benefit from unlabeled samples in the general case if those samples are completely out-of-domain: after all, the core idea of semi-supervised learning is to grasp a better prior on the data domain. I can see that the network can still learn information e.g. when the inputs share the same modality (RGB data) or has an overlap of the classes. Overall, I would make this clearer in the introduction what one expects from semi-supervised learning in an out-of-domain setting. One thing is that semi-supervision should not degrade performance w.r.t. a purely supervised setting, which can happen with current semi-supervised algorithms.

b) I would also experiment with random noise unrelated to the supervised data distribution to see the limits of the approach, and study a case of extreme domain mismatch. In such setting, one would hope to match the purely supervised baseline performance. I expect a batch-norm adaptation to be insufficient for this.

c) I assume that at test-time, the batch norm statistics computed on the supervised set are used; I would make this clear in the document.

I think that adapting batch norm is sufficient in the experiments done but probably not a universal remedy to domain shift in semi-supervised learning, which could be shown with extreme distribution. In general, extra experiments could also show a more progressive evaluation of different shifts, between same-domain unsupervised data, and fully out-of-domain unsupervised data.

I think however that this idea raises some valid points and introduces an easy fix that can be enough in some cases; moreover split-BN can stimulate new ideas related to domain shift and out-of-domain unsupervised learning. Therefore I believe this paper has its place in the workshop.

---

### Official Review · AnonReviewer1 · 2019-04-11
**Simple idea which makes sense and works well in practice**

**Rating:** 3
**Confidence:** 2

**Review:**

Summary:

The authors argue that distribution shift can be detrimental when doing semi-supervised learning. As a simple fix, they propose to not share batchnorm statistics between labeled and unlabeled data. They show consistent improvement for the case where little unlabeled data contains examples for the classes of which labels are present.

Novelty:

The idea to have separate batchnorm parameters seems natural and also seems to work for the problem described here. However, conditional batchnorm is a well-known technique in general so the overall novelty is limited.

Rating:

The overall empirical results are consistent and testing on common perturbations is very insightful. I think the paper is not outstanding, but a simple and valuable contribution to the workshop.

---

### Official Review · AnonReviewer3 · 2019-04-12
**Hypothesis clearly defined and tested**

**Rating:** 4
**Confidence:** 2

**Review:**

This paper improves performance in two modern semi-supervised learning (SSL) models which utilize batch-norm in cases where the models train on unlabeled data from a different distribution than labeled data.  Their simple technique consists of calculating separate statistics for the unlabeled and labeled data in batch-norm.

The paper, which appears to be largely motivated by Section 4.4 of Oliver et al, flushes out the class mismatch problem presented in the aforementioned paper and also tests performance under domain shift. The choices for domain shift perturbations seem reasonable, if not totally realistic.

Although the paper clearly demonstrates improved performance in models with batch-norm, I think that the discussion presented in Section 3.3  warrants additional investigation in future work.

All in all, the paper's hypothesis was clearly defined and tested with thorough experiments and explanation. In addition, the problem definition and proposed solution, though fairly narrow, fits neatly into the workshop format.

Editing comment:
- “which we select 20 randomly classes as the supervised dataset.”

---

### Decision · Program_Chairs · 2019-04-12
**Acceptance Decision**

Accept